# Strategies for the production of isotopically labelled Fab fragments of therapeutic antibodies in *Komagataella phaffii* (*Pichia pastoris*) and *Escherichia coli* for NMR studies

**Donald Gagné**[1], **Muzaddid Sarker**[1], **Geneviève Gingras**[1], **Derek J. Hodgson**[1], **Grant Frahm**[1], **Marybeth Creskey**[1], **Barry Lorbetskie**[1], **Stewart Bigelow**[1], **Jun Wang**[1], **Xu Zhang**[1], **Michael J. W. Johnston**[1,2], **Huixin Lu**[1], **Yves Aubin**[1,2]*

**1** Regulatory Research Division, Center for Oncology, Radiopharmaceuticals and Research, Health Canada, Ottawa, ON, Canada, **2** Department of Chemistry, Carleton University, Ottawa, ON, Canada

* yves.aubin@hc-sc.gc.ca

## Abstract

The importance and fast growth of therapeutic monoclonal antibodies, both innovator and biosimilar products, have triggered the need for the development of characterization methods at high resolution such as nuclear magnetic resonance (NMR) spectroscopy. However, the full power of NMR spectroscopy cannot be unleashed without labelling the mAb of interest with NMR-active isotopes. Here, we present strategies using either *Komagataella phaffii* (*Pichia pastoris*) or *Escherichia coli* that can be widely applied for the production of the antigen-binding fragment (Fab) of therapeutic antibodies of immunoglobulin G1 kappa isotype. The *E. coli* approach consists of expressing Fab fragments as a single polypeptide chain with a cleavable linker between the heavy and light chain in inclusion bodies, while *K. phaffii* secretes a properly folded fragment in the culture media. After optimization, the protocol yielded 10–45 mg of single chain adalimumab-Fab, trastuzumab-Fab, rituximab-Fab, and NISTmAb-Fab per liter of culture. Comparison of the 2D-$^1$H-$^{15}$N-HSQC spectra of each Fab fragment, without their polyhistidine tag and linker, with the corresponding Fab from the innovator product showed that all four fragments have folded into the correct conformation. Production of $^2$H-$^{13}$C-$^{15}$N-adalimumab-scFab and $^2$H-$^{13}$C-$^{15}$N-trastuzumab-scFab (>98% enrichment for all three isotopes) yielded NMR samples where all amide deuterons have completely exchanged back to proton during the refolding procedure.

## Introduction

Therapeutic monoclonal antibodies (mAbs) are the fastest growing class of biotherapeutics. Currently, there are over one hundred innovative mAbs either approved or under review in Canada, Europe and the United States with over forty in the past three years [1]. The majority of them are immunoglobulin of class G and subclass 1 (IgG1) but also includes other subclasses (G2 and G4), chimeric, bi-specific, antibody drug conjugates and fragments. A number

**Data Availability Statement:** All relevant data are within the paper and its Supporting Information files.

**Funding:** My laboratory is part of the department of Health in the Government of Canada, therefore our funding is solely from internal budgets. The funder (i.e. employer, the Government of Canada) had no input, nor influence in the study design, data collection and analysis. and decision to publish nor manuscript preparation.

**Competing interests:** The authors have declared that no competing interests exist.

of these products have lost patent protection and are facing market competition from a growing number of approved biosimilar counterparts, and many are under review by regulatory agencies.

In order to gain market authorization, biosimilars must show biosimilarity with an approved product, the comparator, through a comparability exercise. The latter consists of an array of physico-chemical methods and biological assays to demonstrate biosimilarity which leads to a reduced package of clinical testing. This need to demonstrate biosimilarity has sparked the development of new techniques that were not available to innovators during product development and manufacturing, even though comparability exercises were required by regulatory agencies in cases where manufacturing changes were needed. Amongst the various quality attributes that must be assessed, the higher order structure (HOS) is the most important since it can directly impact potency of the product.

Therapeutic antibodies of the IgG1 subclass are composed of two identical heavy chains and two identical light chains held together with disulfide bonds stabilizing the assembly with a molecular weight of $\sim$150 kDa (Fig 1A). Each heavy chain folds into four immunoglobulin domains ($V_H$, $C_H1$, $C_H2$, $C_H3$) and the light chains fold into two such domains ($V_L$, $C_L$), where V and C refer to variable and constant amino acid sequences. The antigen binding fragment (Fab) is formed with ($V_H$, $C_H1$, $V_L$, $C_L$) while the fragment crystallizable region (Fc) is formed with the $C_H2$ and $C_H3$ from both heavy chains. The antibody binds its target antigen via its Complementarity-Determining Regions (CDRs), located on variable regions $V_L$ and $V_H$ [2]. Each chain contains three CDRs, with all six being involved in the interaction with the antigen [3]. Sequence alignment identifies CDRs as the most variable regions between antibodies (Fig 1A). The Fc fragment, on the other hand, is highly conserved throughout all IgG1 therapeutics [4], which interacts with cell surface receptors.

Recently, nuclear magnetic resonance spectroscopy (NMR) methods have been developed to assess the HOS of mAbs in the context of comparability studies of products [4–7]. The ability of NMR spectroscopy to provide high-resolution information on HOS of mAbs and its fragments at natural abundance was an important achievement. However, these methods could not take advantage of the full power of multi-dimensional (3D and 4D) NMR techniques in order to assign all resonances of all nuclei, to extract detailed structural information, and to study protein dynamics under various formulation conditions. NMR techniques used for the study of large molecular weight proteins, such as Fab or Fc fragments of mAbs (50 kDa), require the incorporation of stable isotopes such as nitrogen-15, carbon-13, and deuterium. While the first two isotopes are needed for the collection of multidimensional data, the latter, deuterium, is needed to replace carbon-bound hydrogens in order to slow down the decay of the carbon-13 magnetization (signal) that is very fast in proteins of the size of Fab fragments (50 kDa). This can be achieved economically only by using either the bacterium *Escherichia coli* or the methylotrophic yeast *Komagataella phaffii* (*Pichia pastoris*). These two organisms can grow on simple sources of nitrogen (ammonium chloride or sulfate) and carbon (glucose, glycerol, or methanol) that can be substituted with their labelled counterparts. In addition, only *E. coli* and *K. phaffii* can tolerate the high levels (85–99%) of deuterium oxide ($D_2O$) used as a substitute for water in culture media and still produce multi-milligram quantities of target proteins required for NMR studies. While *E. coli* have some advantages, such as simpler molecular biology procedures, faster growth and protein expression, over the yeast *K. phaffii*, the latter is better suited to express soluble multi-domains and multi-chain proteins containing disulfide bridges encountered in mAbs, or their Fab and Fc fragments, which result from the assemblage of two polypeptide chains that are stabilized by intra-domain and inter-chain disulfide bonds. *Escherichia coli* lacks the cellular machinery for the formation of proper disulfide bonds [8]. Nevertheless, several approaches were developed to improve refolding in

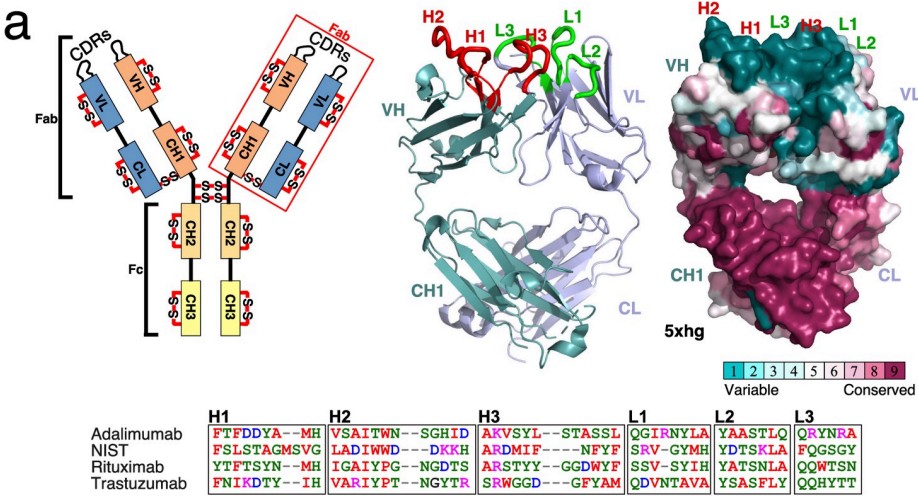

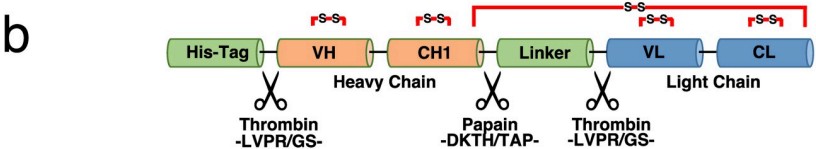

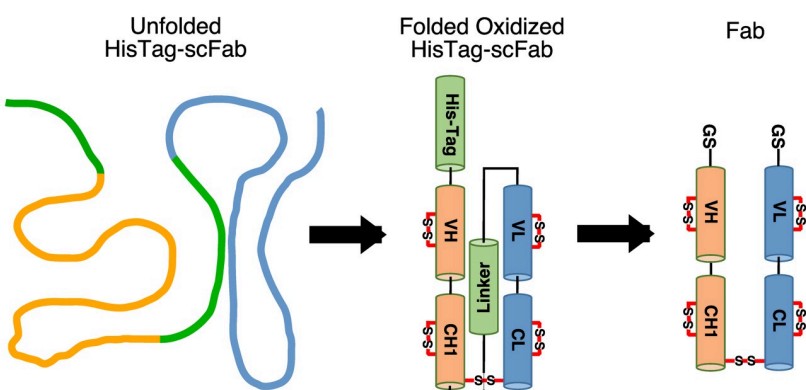

**Fig 1. Design of the single chain Fab constructs.** A) Structural elements of a monoclonal antibody and sequence variants of the four mAbs produced. B) Schematic representation of the construct and production of the final Fab.

bacteria, including the addition of fusion proteins, protein expression at lower temperature, the inclusion of a signal peptide for refolding in the periplasm (by the periplasmic oxidoreductase DsbA), and more. In most cases, yields remain low when they are reported [9–13], which may become abysmally low when attempting to produce fully deuterated proteins. In addition, the yeast *K. phaffii* has the ability to produce N-glycosylated Fc fragments [14], which are different from N-glycans in humans.

Here, we present our strategies for the preparation of isotopically labelled ($^2$H, $^{13}$C, $^{15}$N) Fab fragments using *K. phaffii* and *E. coli*. Initially, we developed a *K. phaffii* expression system based on the sequence of the NISTmAb RM8671 reference material. The isotopically labelled Fab fragment produced with this expression system was used to support the NIST collaborative study reported earlier [7]. Expression of soluble triply labelled Fab fragments has two disadvantages. Soluble protein expression in 90% (and above) deuterium oxide growth media significantly decreases yields, and the resulting protein produces NMR spectra where many backbone amide resonances have either low or no intensity at all. This is due to the fact that the immunoglobulin fold is very stable and deuterium to hydrogen exchange can be very slow for many amides (days to months), which requires denaturation of the Fab to accelerate the exchange in order to recover all amide protons [15]. As an alternative, we have developed an approach based on the *E. coli* expression of a single polypeptide chain with a removable linker between the heavy and light chains in order to facilitate refolding from high-level expression of inclusion bodies (see Fig 1B). The presence of the linker ensures a 1:1 heavy chain to light chain stoichiometric ratio thus facilitating proper refolding. In addition, refolding a triply labelled ($^2$H, $^{13}$C, $^{15}$N) polypeptide produces a target protein with all exchangeable backbone amides fully protonated. For these reasons, we aimed directly for the production of inclusion bodies. Buchner and Rudolph [16] and Gani and coworkers [17] have reported refolding of separately expressed heavy and light chains, however, we did not consider this approach. Instead, we extended the concept applied to single chain Fragment variable (scFv) where the variable domains ($V_H$ and $V_L$) of an antibody are linked with a polypeptide linker and expressed as a single chain. This concept had been tested for the soluble expression of Fab fragments [11, 18–20], however, we added the capability to remove the linker by taking advantage of the papain cleaving site in the hinge region of the heavy chain and by adding a thrombin cleaving site at the amino-terminal end of the light chain. In addition, a cleavable polyhistidine tag was added to the amino-terminal end of the heavy chain to allow on-column refolding [21] among other refolding methods. This single chain strategy was tested on four Fab fragments: adalimumab-Fab (Humira®), NISTmAb-Fab (RM 8671), rituximab-Fab (Rituxan®), and trastuzumab-Fab (Herceptin®). The use of thrombin to remove the tag and linker at the amino-terminal ends would leave only serine-glycine dipeptides that would induce negligible differences on NMR spectra of the final fragment compared to the Fab obtained from the papain cleavage of corresponding commercial drugs.

## Material and methods

### Expression system of the NISTmAb in K. phaffii and production of isotopically labelled NISTmAb-Fab

**Plasmid expression system for NISTmAb-Fab.** Initially, sequences corresponding to the heavy and light chain of the NIST reference standard monoclonal antibody RM 8671 (https://www.nist.gov/programs-projects/nist-monoclonal-antibody-reference-material-8671), were synthesized (BioBasic) for optimal expression in the methylotrophic yeast *K. phaffii*. Sequences included a *Kex2* gene product as well as *Ste13* gene product recognition sites (Glu-Lys-Arg-Glu-Ala-Glu-Ala) in N-terminal for further cleavage of the secretion signal, namely the native *Saccharomyces cerevisiae* alpha factor secretion signal. Sequences were each inserted separately into a vector pPICZalphaA (Invitrogen) using the *XbaI/XhoI* restriction sites, creating respectively construct A and construct B (see S1 Table for all constructs). A bicistronic vector was then created, as described by Schoonooghe and coworkers [22]. Briefly, the *PmeI* restriction site from construct B was removed by site-directed mutagenesis (Stratagene) using primers 1 and 2 (see S2 Table for all primers) to create construct C. Vectors were then digested, construct

A with *PciI and BamHI* and construct C with *BglII* and *PciI*, and re-ligated together to form the bicistronic vector construct D.

The NISTmAb-Fab was constructed by inserting a stop codon in the hinge region of the above plasmid using site-directed mutagenesis with primers 3 and 4 of the Fab fragment of the NIST mAb with construct E.

**Protein expression.** After linearization with *Pme1*, the various vectors were electroporated into *K. phaffii* (X-33 and/or KM-71H strains) according to standard protocols using a BioRad Gene Pulser Xcell Electroporation System. Following the manufacturer's protocol, (Pichia Expression kit manual ThermoFisher Scientific, cat# K171001), clones were selected based on secreted expression after growth at 29ºC in buffered minimal glycerol (BMG) at pH 6.0 (100 mM potassium phosphate, 3.4 g/L yeast nitrogen base (YNB) (Sigma), 10 g/L ammonium sulfate, 10mL/L glycerol and biotin 4 μg/L) and induction in buffered minimal methanol (BMM) at pH 6.0 (100 mM potassium phosphate, 3.4 g/L yeast nitrogen base (Millipore-Sigma), 10 g/L ammonium sulfate, 5 mL/L methanol and 4 μg/L biotin) with fresh 5 mL/L methanol supplementation every 24h. Expression of the secreted protein of interest was detected by SDS-PAGE and confirmed by mass spectrometry analysis.

**Expression and growth media for isotopic labelling with $^{15}$N and $^{13}$C in flasks.** A single colony was grown at 29ºC in BMG (100 mM potassium phosphate pH = 6.0, 3.4 g/L YNB, 10 g/L (NH$_4$)$_2$SO$_4$, 20 g/L (2% carbon source) glucose and 0.4 mg/L biotin) until OD$_{600}$ reached between 1 and 6. Cells were then harvested (2500 x g, 5 min at RT) and washed in 1/10 volume of medium without a carbon source for 30–60 min prior to being re-suspended in BMM (100 mM potassium phosphate pH 6.0, 3.4 g/L YNB, 10g/L (NH$_4$)$_2$SO$_4$, 5 mL/L methanol and 0.4 mg/L biotin) to an OD$_{600}$ of 1. Induction was carried out at 29ºC, with fresh methanol (5 mL/L) supplementing the media every 24h. Ratios of carbon-13 and nitrogen-15 for isotopic labelling were determined as required.

**Expression and growth media for isotopic labelling with $^{15}$N and $^{13}$C in bioreactor.** A single colony was inoculated in 100 mL of buffered minimal glycerol (or glucose) (100 mM potassium phosphate pH 6.0, 3.4 g/L YNB, 10 g/L (NH$_4$)$_2$SO$_4$, 10 g/L (1% carbon source) of glycerol or glucose and 0.4 mg/L biotin) and grown at 29ºC for 24 h. This starter culture was diluted in 1.5 L of buffered minimal glycerol (or glucose) in a 5 L bioreactor with pH and dissolved oxygen (DO) control using 2N KOH for pH and air and agitation for DO. Set points were at pH 6.0 and at DO 30%. The starting OD$_{600}$ was 1.0 and the culture was incubated for 25 h until the OD$_{600}$ reached 20 (OD$_{600}$ = 1.0 after 20X dilution). Protein synthesis was induced by continuous addition of 50% methanol at a feed rate of ∼ 10 mL/h for 5 min, then slowed at ∼ 2 mL/h for 48 h.

**Purification of NISTmAb-Fab.** At 65-70h post-induction, cells were harvested. Media and cells were separated by centrifugation (3000 x g, 10 min at 4ºC). The pH of the supernatant was adjusted to 7.3 with 10N KOH after addition of 0.5mM -1 mM sodium ethylenediamine-tetraacetate (EDTA) and 5mM benzamidine as proteases inhibitor. Upon addition of KOH, a white precipitate formed into the otherwise clear solution and was let to settle for about 30 min after which the solution was centrifuged again (3000 x g, 10 min) prior to affinity chromatography. Purification of the NISTmAb-Fab was carried out by loading solution on a Capture-Select$^{TM}$ IgG-CH1 affinity matrix (Life Technologies) column using 5mL of resin per liter of culture supernatant at a flow rate of 0.5 mL/min. After loading, the column was washed with 10 column volumes (CV) of buffer A (1x PBS: 8g/L (136.9 mM) NaCl, 0.2g (2.7 mM) KCl, 1.44g/L (10.1 mM) Na$_2$HPO$_4$, 0.24g/L (1.8 mM) KH$_2$PO$_4$ with 5 mM benzamidine and 0.5–1 mM EDTA, at pH 7.3) prior to elution with buffer B (0.1 M Glycine pH 3.0). Fractions containing the NISTmAb-Fab were pooled and the buffer was exchanged and concentrated using a 10 kDa MWCO ultracentrifugation filter and purified by size exclusion chromatography

using a HiLoad-26/60 Superdex 75 (GE) with buffer C (50 mM Acetate, 150 mM NaCl, 5 mM benzamidine, 0.5 mM EDTA, pH 5.7). Fractions were pooled, buffer exchanged and concentrated to 2.5 mg/mL in 25 mM bis-Tris d$_{19}$, pH 6.0, or 50 mM sodium acetate-d$_3$ using a 10 kDa MWCO centrifugal filter for NMR analysis.

**Adaptation of K. phaffii cells to D$_2$O.** Production of carbon-bound deuterium samples was carried out using two methods of deuteration. In the first method, cells harboring construct D for the expression of the NISTmAb-Fab were acclimated first in buffered minimal glucose media with 70% D$_2$O before increasing deuterium to 99%, as described by Pickford et. al. [23] and Morgan et.al. [24], prior to induction with methanol at a low and high cell density (0.7 and 6.0 respectively) in a buffered minimal methanol media for 96 h, as described earlier. In the second method, cells for the expression of the NISTmAb-Fab were slowly adapted to D$_2$O. In BMG, the concentration in D$_2$O was increased stepwise from 0%; 25%; 50%; 70%; 85%; 90%, and finally 99% of deuterated water. Three passages of *K. phaffii* culture were carried out at each deuterium enrichment level when cell density was between OD$_{600}$ 1.0–6.0. Inoculation was always carried out at a cell density of 0.1 at each passage or change of enrichment level.

## Expression in *Escherichia coli* and production of isotopically labelled Fab fragment of mAbs

**Construction of scFab expression vectors.** DNA sequences for heavy and light chains of adalimumab, rituximab, and trastuzumab were extracted from DrugBank (https://go.drug-bank.com), while the NISTmAb DNA sequence was obtained from Formolo and coworkers [25]. The genes for the single chain Fab fragments were constructed in the following manner: a ten-histidine tag with sequence MSGSS HHHHH HHHHH SSGHM, followed by a thrombin protease cleavage site LVPRGS, was fused to the sequence of the heavy chain of the target Fab. Since all target mAbs are papain cleavable at the hinge region between the histidine and threonine (in bold), the linker was fused after the proline (CDKT**H**-**T**C(A)P) where the cysteine was mutated to an alanine in bracket. A simple 30-residue linker was made of 6 repeats of the GGGGS sequence followed by SSSG and a thrombin cleavage site just before the first residues of the light chain (Fig 1). DNA sequences were codon optimized for expression in *E. coli*, and synthesized by BioBasic Inc. (Markham ON Canada). The synthetic DNA was inserted into a pET15b expression vector (Novagen) at the *Nco*I and *Bam*HI sites. The quality of each construct and absence of mutations was verified by gel and gene sequencing.

**Protein expression.** A single freshly transformed colony of BL21(DE3) with the target plasmid was incubated in 2 mL of Luria-Bertani (LB) medium containing ampicillin (100 μg/mL) at 37˚C. After 6–7 h, 100 μL of this culture was diluted into 100 mL of M9 media containing 1 g/L of $^{15}$N-ammonium chloride at pH 7.4, 0.1 μM of calcium chloride, 1 μM of magnesium sulfate, supplemented with 1X MEM Vitamin Solution, 3 g/L of glucose (or 2 g/L $^{13}$C-U6-glucose) and 100 μg/mL of ampicillin, and incubated overnight at 37˚C. The next day, this inoculum was diluted into 900 mL of M9 media and incubated at 37˚C. Protein synthesis was induced with the addition of 1mM IPTG once OD$_{600}$ reached 0.8. Induction phase was left to proceed for up to 24 h at 37˚C depending on the fragment.

Expression of $^{2}$H-$^{13}$C-$^{15}$N-histag-scFab of adalimumab and trastuzumab was carried out as described above with the following modifications. A freshly transformed colony was incubated in 5 mL of LB media with 100 μg/mL of ampicillin for 2–3 h at 37˚C. A 100 μL aliquot of this culture was diluted in 50 mL M9 in H$_2$O and incubated 5–6 h at 37˚C. Once OD$_{600}$ reached 0.25, 1 mL of this inoculum was diluted in 100 mL of M9-D$_2$O and incubated at 37˚C overnight (16 h). The M9-D$_2$O contained 1 g/L of NH$_4$Cl, and 2 g/L of U-$^{13}$C6-Glucose

1,2,3,4,5,6,6-d7. All solutions ($CaCl_2$, $MgSO_4$, ampicillin and IPTG) were prepared with deuterium oxide instead of water. The overnight culture was further diluted in 1 L of M9-$D_2O$. Protein expression was induced at $OD_{600}$ of 0.7. Protein synthesis was allowed to proceed for 21–23 h. Cells were harvested at $OD_{600}$ of >2.0 by centrifugation at 3,000 x g for 30 min, and the pellet was stored at -80˚C.

**On-column protein refolding.** Cell pellets (1.5–2.0 g) were resuspended into 35 mL of a buffer G (10 mM Tris-HCl and 100 mM sodium phosphate at pH 8.0, 6 M guanidium chloride (GdmCl), and 20 mM of freshly added reduced glutathione). Cells were sonicated using a Branson model 450 sonifier with a microtip with 5 s pulses at 40% power spaced by 5 min delays. Sonication was repeated a second time after cooling to 4˚C for 10–15 min. Cell debris were pelleted down by centrifugation at 17,000 x g for 30 min. The supernatant was set aside, and the pellet was resuspended in another 35 mL of buffer G. Sonication and centrifugation were repeated a second time. Supernatants were combined and incubated with 20 mL of washed nickel-NTA (Ni-NTA) resin with nutation for 60 min at room temperature. The mixture was loaded into a glass econo-column® (Bio-Rad, CA United States). The resin was washed with 5 column volumes (CV) of buffer G with 10 mM Imidazole, followed by 5 CV of buffer G with 25 mM imidazole to removed non-specific protein binding. Buffer G was added to the resin and the mixture transferred to a XK-26 empty glass column (Cytiva, MA, United States), then mounted to the FPLC (Bio-Rad, CA, United States). A cycle starts with a linear gradient from 100% to 50% of buffer G against buffer B (10 mM Tris-HCl and 100 mM phosphate, pH 8.0) at 2 mL/min over 10 CV, followed by a second linear gradient from 50% to 0% of buffer G against buffer B at 1 mL/min over 10 CV. The resin was washed with buffer B, and polyhistidine tag-containing proteins were eluted with 250 mM imidazole over 5 CV at 2 mL/min. The cycle was repeated until there was no elution peak present.

**Drop-by-drop fast dilution protein refolding.** The cell pellet from a one-liter culture was reconstituted with 35 mL of lysis buffer (20 mM Tris-HCl, 2.5 mM DTT, pH 8.5), and sonicated as described earlier. Inclusion bodies were recovered by centrifugation at 30,000 x g for 30 min (4˚C). The pellet was resuspended by agitation in 20 mL of buffer (100 mM Tris-HCl; 6 M GdmCl; 2 mM EDTA; 80 mM reduced glutathione; 2.5 mM DTT, pH 8.5) for 2–3 h at 4˚C. The resuspended pellet was added drop-by-drop, with frequent pauses, to the buffer RB (20 mM Tris-HCl; 500 mM L-arginine; 3.2 mM oxidized glutathione; pH 9.0) while stirring, and frequent pause of 30–60 min were taken during the addition. Alternatively, a syringe pump set at 30 mL/h was used to slowly add the resuspended pellet in the buffer RB. The volume of the refolding buffer was 2 L in order to keep protein density at 100 μg/L or less to minimize intermolecular interaction in order to maximize refolding efficiency with an oxidized to reduced glutathione ratio of ≥4. Refolding solution was stirred gently at 4˚C for 65–70 h.

**Dialysis slow dilution protein refolding.** After lysis, inclusion bodies corresponding to a 0.5 L culture were resuspended in 50 mL of denaturation buffer DN (100 mM Tris-acetate, 2 mM EDTA, 6 M GdmCl, 100 mM reduced glutathione, pH 9) under vigorous stirring for 2 h at 4˚C. This solution is then diluted with 50 mL of buffer DN supplemented with 1M L-arginine. This solution is then dialysed against 400 mL of buffer DL (100 mM Tris-acetate, 2 mM EDTA, 500 mM L-arginine, pH 9.0) with 2.25 M GdmCl and 50 mM reduced glutathione overnight (15–16 h) at 4˚C. The second dialysis was carried out against 500 mL of buffer DL with 0.5 M GdmCl and 25 mM reduced glutathione for 8 h at 4˚C, and the third dialysis was carried out against 500 mL of buffer DL with 12.5 mM reduced glutathione overnight at 4˚C. The final dialysis was performed against 500 mL of buffer DL with 5 mM oxidized glutathione for 6 h at 4˚C. Prior to purification with CaptureSelect™ resin, the content of the dialysis bag (∼125 mL) was diluted with 900 mL of 100 mM Tris-acetate pH 6.0 to obtain a solution at pH 7.0–7.2.

**Purification from drop-by-drop and dialysis refolding.** The refolding solution was dialyzed three times against 20 mM Tris-HCl at pH 6.5–7.0 (for CaptureSelect™), or pH 7.5–8.0 (for NiNTA) with a ratio of 1:8 (refolding:buffer) using SnakeSkin dialysis bags with 10 kDa MWCO (ThermoFisher). The pH of the dialysate was then verified and adjusted if required. The dialysate was centrifuged at 3,000 x g for 30 min at 4˚C to remove debris. The solution was then filtered on 5 μm filter before loading onto the column.

The dialysate was loaded at 2 mL/min on a 10 mL of resin NiNTA (Qiagen) in a XK 26/20 (Cytiva) column pre-equilibrated with 20 mM Tris-HCl at pH 8.0. Unbound impurities were washed with 5 CV of loading buffer. Polyhistidine tag containing fragments were eluted with 20 mM Tris-HCl and 250 mM imidazole at pH 8.0. Sodium ethylenediaminetetraacetate (2 mM) was added to each fraction to chelate any nickel leachate. Fractions containing the target protein were pooled, concentrated up to 3–5 mg/mL, and buffer exchanged using Amicon Ultra-15 and Amicon 0.5 ultrafiltration devices with 10 kDa MWCO, (EMD-Millipore). Alternatively, the dialysate was loaded on a 10 mL CaptureSelect™ (Life Technology, CA United States) pre-equilibrated with 20 mM Tris-HCl pH 7.3 buffer at 4 mL/min. Proteins were eluted with 10 CV of 500 mM acetic acid, followed by another 10 CV of 100 mM glycine-HCl buffer at pH 2.5. Fractions containing the target protein were pooled and concentrated to 3–5 mg/ mL using Amicon-Ultra-15.

**Protease cleavage with thrombin and papain.** Proteins were buffer exchanged by dialysis in 100 mM sodium phosphate, pH 7.0. The sample was further centrifuged at 12,000 x g for 10 min to remove any precipitated proteins. Thrombin cleavage of the polyhistidine tag was carried out with 25 U of thrombin (Cytiva, Ref. 27084601) per milligram of protein in 20 mM sodium phosphate, pH 7.0 at 37˚C. Cleavage of the polyhistidine tag was completed after 5 h. Papain was sourced from Papaya Latex (MilliporeSigma, ON Canada) and used at a 1:100–400 ratio (protein:papain) in papain cleavage buffer (20 mM sodium phosphate; 5 mM EDTA; pH 7.0) supplemented with 3.5 mg/mL of L-cysteine-HCl. The cleavage was conducted at 37˚C for 16 h, with light agitation (75–100 rpm). The solution was diluted 10-fold in 20 mM sodium acetate at pH 5.0, and further purified with cation exchange chromatography using a 5 mL HiTrap SP FF prepacked column (Cytiva) with a 0–0.5 M linear gradient of sodium chloride. The target protein eluted around 250 mM sodium chloride while papain eluted at 400 mM. Fractions containing the Fab fragment were pooled and concentrated. Further purification was carried out using size-exclusion chromatography on a HiLoad 26/60 Superdex 75pg Prep Grade (Cytiva) at a flow rate of 1.5 mL/min with 20 mM sodium acetate pH 5.0 as mobile phase. Fractions containing the protein were pooled, concentrated, and buffer exchanged in 20 mM sodium acetate-d3 at pH 5.0 for analysis.

**Characterization of Fab fragments.** *Preparation of Fab fragments from marketplace products.* Adalimumab (Humira®), Rituximab (Rituxan®), and Trastuzumab (Herceptin®) were purchased from a local pharmacy. Fab fragments were prepared as described previously[4]. NIST (Fab) was provided by Robert Brinson and John P. Marino of the National Institute of Standards and Technologies (Maryland, United-States). Fab fragments were buffer exchanged in 20 mM sodium acetate-d3 at pH 5.0 and kept at 4˚C until used.

*NMR spectroscopy.* Samples were prepared in 20 mM sodium acetate-d3 at pH 5.0 with 5% (v/v) deuterium oxide for lock frequency purposes and transferred to 5 mm tubes. Protein concentrations varied between 0.3–2.7 mg/mL. All experiments were carried out at 50˚C (323 K) on AVANCE III 600 and AVANCE III-HD 700 MHz NMR spectrometers equipped with TCI cryoprobes. Two-dimensional spectra were acquired using a $^1$H-$^{15}$N SOFAST-HMQC [26] experiment with spectral width of 16 (35) ppm with the carrier set at 7.4 (120) ppm in $^1$H ($^{15}$N). Two-dimensional sensitivity enhanced $^1$H-$^{15}$N HSQC experiments with gradient coherence selection were recorded with spectral width of 16 (32) ppm with the carrier set at 7.4

(120) ppm in $^1H$ ($^{15}N$) [27]. Datasets contained 1024 (64) complex points with acquisition time of 92 ms (28 ms) for $^1H$ ($^{15}N$). Methyl spectra were acquired with $^1H$-$^{13}C$ HSQC experiments. Unless specified otherwise, a spectral width of 16 (40) ppm with the carrier set at 7.4 (20) ppm in $^1H$ ($^{13}C$) were used, with 1024 (32) complex points. All NMR spectra were processed with nmrPipe [28] and visualized with NMRViewJ [29].

*Size exclusion chromatography*. All chemicals used for HPLC assays were analytical reagent grade. Potassium chloride, potassium phosphate monobasic and potassium phosphate dibasic were purchased from Millipore Sigma (St. Louis, MO, USA). Gel Filtration Standard (thyroglobulin, bovine gamma-globulin, chicken ovalbumin, equine myoglobin and vitamin B12; MW 1350–670 000) was obtained from Bio-Rad. Distilled water was deionized on a Milli-QTM EQ 7000 water purification system (Millipore SAS, 67120 Molsheim, France). The HPLC system consisted of a Waters Alliance 2695 chromatograph equipped with a column heater and an auto-sampler with a sample cooling device. Fluorescence and UV spectroscopy detection was accomplished by coupling in-series a Waters 2475 Multichannel Fluorescence Detector (excitation wavelength of $\lambda_{ex}$ 280 nm and emission wavelength at $\lambda_{em}$ 335 nm) and a Waters 2998 UV-VIS photodiode array detector with monitoring at 214 nm and 280 nm (Waters, QC, Canada). Data acquisition and integration were performed with Empower 3 Chromatography Data Software. The chromatographic column used was a TSKgel G2000SWXL, 300 mm x 7.8 mm, 5 μm particles, 125 Å pore size (Tosoh Bioscience, Philadelphia, PA, USA). Separations were carried out at room temperature with isocratic elution at 0.5 mL/min using a potassium phosphate buffer, pH 6.2.

*Mass spectrometry*. Samples were analyzed with an Orbitrap Fusion Tribrid Mass Spectrometer coupled to an Easy-nLC 1200 (Thermo Scientific) which was calibrated by infusion prior to analysis with a mixture of caffeine, MRFA, and Ultramark 1621. For LC-MSMS, a 1 μL aliquot was analyzed by loading onto a pepmap C4 trap column (5 mm) and desalting with 0.1% formic acid in water (solvent A) before separating on a ProSwift RP-4H HPLC C4 reverse-phase analytical monolithic column (1mm x 50mm). Chromatographic separation was achieved at a flow rate of 2 μL/min over 23 min in eight linear steps as follows (solvent B was 0.1% formic acid in 80% acetonitrile): initial, 3% B; 10 min, 25% B; 15 min, 50% B; 17 min, 95% B; 19 min, 95% B; 21 min, 3% B; 23 min, 3% B. The eluting proteins were analyzed in Intact Protein/Low Pressure Mode. MS1 was performed in the Orbitrap at a resolution of 15000, scan range 600–3500 m/z, 100 ms maximum injection time, AGC target of 4e5, and 5 microscans. Data processing was carried out with Biopharmafinder 3.0 software package using the deconvolution algorithm ReSpect. Parameters were Sliding Windows with 25% offset, deconvolution mass tolerance 20 ppm, model mass range 10000–70000, charge state range 3–100, minimum adjacent charges 3–10.

*Thermal unfolding monitored by circular dichroism*. Samples were prepared to the appropriate concentrations (range of 0.15–0.9 mg/mL) of protein with a final volume of 400 μL per sample. Sample analyses were run in a 1 mm path length quartz Suprasil cuvette (Hellma, Mullheim, Germany) on a Jasco J-1500 spectropolarimeter (Jasco International Co. Ltd., Tokyo, Japan) equipped with a Peltier thermal control unit. Both the instrument and thermal control unit were controlled using Jasco Spectra Manager Software. Sample analyses included spectral scans at 20˚C (175–280 nm, 5 accumulations) and thermal denaturation between 20 and 90˚C at a rate of 1˚C/min, while monitoring Far-UV CD signal (in millidegrees) at 215 nm, with a data pitch of 1 nm and response time of 1 s. The $T_m$ data were recorded every 1˚C, and a spectral scan (175-280nm) was acquired every 5˚C as well. All spectra were corrected for buffer signal where appropriate. $T_m$ values (the temperature point where 50% of the total change occurs) were estimated from thermal unfolding data at 215 nm.

*Determination of isoelectric point by Capillary IsoElectric Focusing (cIEF)*. All chemicals used for CE assays were analytical reagent grade. L-arginine, iminodiacetic acid, urea, sodium hydroxide and phosphoric acid were purchased from MilliporeSigma (St. Louis, MO, USA). Glacial acetic acid and Pharmalyte 3–10 ampholytes (Cytiva) were purchased from Fisher Scientific (Ottawa, ON, CA). The Advanced cIEF Starter Kit (PN A58481) containing pI markers (4.1, 5.5, 7.0, 9.5, and 10) and cIEF gel buffer were purchased from Sciex (Brea, CA, USA).

All cIEF experiments were performed using a Beckman Coulter (Sciex) PA800 Plus Pharmaceutical Analysis System equipped with a UV detector monitoring at 280 nm. Separations were carried out at 20˚C using a 50 μm I.D. Neutral Capillary (Sciex, PN 477441) with a 20 cm effective length (30.2 cm total length). cIEF solutions were prepared according to the manufacturer's instructions (see https://sciex.com/content/dam/SCIEX/pdf/customer-docs/application-guide/ PA800pluscIEFApplicationGuide.pdf), which included an anolyte (200 mM phosphoric acid), catholyte (300 mM sodium hydroxide), chemical mobilizer (350 mM acetic acid), anodic stabilizer (200 mM iminodiacetic acid), cathodic stabilizer (500 mM L-arginine), capillary cleaning solution (4.3 M urea) and urea-gel for separation (3 M urea in cIEF gel buffer). Samples for cIEF analysis were prepared by mixing 5 μL of Fab (∼2 mg/mL) with 120 μL of cIEF master mix (100 μL cIEF gel buffer (3 M urea); 10 μL Cathodic stabilizer; 1 μL Anodic stabilizer; 6 μL 3–10 ampholytes; 1 μL of each three pI markers). The three pI markers were chosen so that marker peaks did not overlap with Fab peaks. Samples were then vortexed on maximum speed (10 s) and briefly centrifuged (10 s) before transferring to CE vials for analysis.

A standard cIEF method designed by the manufacturer was used (see https://sciex.com/ content /dam/SCIEX/pdf/tech-notes/all/High-res-cIEF.pdf). B riefly, the samples prepared in master mix were injected at 25 psi for 100 s and focused at 25 kV for 15 min. Peaks were chemically mobilized past the detector at 30 kV for 30 min before a water rinse at 50 psi for 2 min. The capillary was rinsed between samples with cleaning solution at 50 psi for 3 min and water at 50 psi for 2 min. A weighted average pI of each Fab was calculated based on the calibration of the three internal pI markers.

## Results

### Production of the NISTmAb-Fab in yeast

The methylotrophic yeast *Komagataella phaffii* (*Pichia pastoris*) was chosen to produce a Fab fragment derived from the NISTmAb antibody for its ability to incorporate stable isotopes and grow in high concentrations of deuterium oxide. The target protein, NISTmAb-Fab, was secreted in the culture milieu. Initially, the NISTmAb-Fab was isolated from the media using cation exchange chromatography at pH 5.0. A quick NMR analysis and subsequent confirmation by mass spectrometry of this crude product showed that a high proportion of the sample (>50–60%) was constituted by the light chain and the remainder being the Fab. Affinity purification using CaptureSelect$^{TM}$ IgG-CH1 affinity matrix yielded the NISTmAb-Fab (S1 Fig). Overlay of the NMR spectrum of the yeast-produced Fab and the NISTmAb-Fab RM8761 Fab at natural abundance showed that almost all resonances have a near perfect match indicating that the yeast produced material has the same three-dimensional structure as the reference material. The extra resonances result from the tetratpeptide (Glu-Ala-Glu-Ala) at both N-terminal ends of the heavy and light chains that remain after cleavage of the alpha factor secretion signal.

After two purification steps, protein yields after were 2–3 mg per liter from flasks culture. Attempts to produce the fragment in bioreactor yielded 6 mg/L when using 1% glycerol as carbon source and 4 mg/L when using 1% glucose. Yields from soluble protein expression of the NIST-mAb-Fab of 2–6 mg/L of *K. phaffii* cultures correspond to 0.2 mg of protein per gram of carbon source, excluding methanol used to induce protein expression that contribute to about

40% isotope incorporation. Production of a doubly labelled $^2$H-$^{15}$N-NISTmAb-Fab was carried out using glucose as the carbon source (prior to protein induction) because of its lower cost. In our hands, protein yields were significantly lower (c.a. <40%) with deuterium oxide being the sole deuterium source. This suggested that aiming for 98% deuteration using doubly labelled glucose (99%) would be too costly. Considering the reduced yields of deuterated NISTmAb-Fab, the cost of isotopes, and the need to eventually unfold the labelled protein to fully exchange amide deuterons to protons, we elected to develop a method using *E. coli*.

## Production of Fab fragments in bacteria

Expression of histag-scFab of the NIST-mAb, adalimumab, rituximab, and trastuzumab was initially optimized by testing induction temperature (16 and 37°C) and duration (3-24h). All four constructs expressed best at 37°C while duration varied between 18–24 hours depending on the Fab. While rituximab required longer expression time (see S2 Fig), all fragments produced large amounts of inclusion bodies (c.a. 114–140 mg per liter of culture measured in 6 M guanidinium chloride, see Table 1).

## On-column refolding

This procedure consists of immobilizing the target protein on a nickel affinity resin to prevent or minimize intermolecular interactions leading to the formation of protein aggregates. The denaturant is removed via a gradient (linear or else) with the same buffer without denaturant under oxidative conditions prior to elution of the protein. The procedure can be repeated by re-equilibrating the column with the denaturing buffer, then the cycle of denaturant removal via a gradient and protein elution is repeated to obtain additional amounts of folded protein. This method has been used successfully for the high-yield production of cytokines [30–32] and the NISTmAb-scFv fragment [21]. While three cycles of refolding were needed for cytokines, the NISTmAb-scFv required up to 10 cycles. The decision on the number of cycles needed is based on the intensity of the eluted peaks. In all cases, well-folded proteins were obtained after elution off the column. All these proteins contained two disulfide bridges. Initial attempts to refold the histag-NISTmAb-scFab (S3 Fig) suggested that the refolding of a single chain constructs seemed possible. A proof-of-concept was emerging when this protocol was applied to the other Fab fragments, which produced low amounts of well folded Fab fragments. However, large amounts of soluble but misfolded protein also eluted off the column. In fact, any manipulation of the eluate (e.g., buffer exchange, protease cleavage) led to significant (> c.a. 70–90%) loss of material. Attempts at optimizing a few parameters, such as column size to dilute unfolded scFab to minimize intermolecular interactions, gradient slope of denaturant removal, and flow rate, showed no improvement in yields of refolded Fab. This mix of folded and misfolded soluble species in the eluate was initially puzzling considering that only correctly folded cytokines and NISTmAb-scFv eluted off the column at any given cycle. Note that yields were simply estimated from 2D-$^1$H-$^{15}$H-NMR spectra, which were used to ensure that Fab

**Table 1. Yields of inclusion bodies (measured by absorbance in 6M GdmCl) per liter of *E. coli* culture and per quantity of pelleted cells.**

| Histag-mAb-scFab | Yield (mg/L) | Yield (mg/g) |
|---|---|---|
| adalimumab | 120.9 | 30.2 |
| NIST-mAb | 138.3 | 27.3 |
| rituximab | 114.0 | 17.8 |
| trastuzumab | 123.5 | 51.4 |

fragments displayed a properly folded conformation by comparing with the corresponding NMR spectrum of the Fab from the parent drug [4]. Trastuzumab-scFab gave largest yield (c.a. <1mg per liter of culture), followed by the NISTmAb-scFab and adalimumab-scFab and no detectable amount of rituximab-scFab. Based on these results, we concluded that scFab fragments may have a complex refolding pathway that on-column refolding may not be able to accommodate. Therefore, we elected to test other refolding methods that can provide more degrees of freedom in terms of refolding conditions not allowed in the context of a nickel immobilized resin.

## Drop-by-drop fast dilution refolding method

The initial attempt of the fast dilution method to refold a histag-scFab produced from *E. coli* inclusion bodies used similar conditions as described by Boix and coworkers [33]. Typically, inclusion bodies recovered after cell lysis of one-liter culture of *E. coli* were dissolved in 20 mL of 6 M guanidium chloride under reducing conditions (80 mM reduced glutathione) and added drop-by-drop to a large volume (100x) of refolding solution containing 0.5 M of L-arginine and 3.2 mM oxidized glutathione (GSSG) (to keep a 1:4 ratio of GSH:GSSG) under moderate stirring in 20 mM Tris-HCl buffer pH 8.5. With these conditions, adalimumab-scFab showed higher yield of refolded protein, compared to previous attempts, and rituximab showed promising results, namely well folded fragments in the presence of other species as determined by 2D-NMR. In light of these results, we set out to optimize refolding conditions summarized in Table 2.

Using the selected conditions, the four histag-scFab (histag-NISTmAb-scFab, histag-adalimumab-scFab, histag-rituximab-scFab, and histag-trastuzumab-scFab) were prepared, obtaining yields of 10–45 mg per liter of culture, after the first purification step. Papain treatment of these single chain fragments produced Fab fragments for final characterization.

## Dialysis slow dilution refolding method

The refolding method by stepwise dialysis showed that adalimumab can be refolded with equal or higher yields, while all other showed lower yields (not accurately quantified). Histag-adalimumab-scFab refolded with the fast dilution method gave 8–10 mg/L, while refolding by dialysis produced 11 mg/L. The expected advantage of the dialysis method was to keep the volume of sample as low as possible to speed loading on the affinity column. However, dialysis against buffer suitable to the CaptureSelect™ purification step to remove arginine, traces of denaturant, and reducing agent led to less than half the yield (4 mg/L). This is likely the result of precipitation of unstable species that may also bring down some properly folded histag-scFab. The 11 mg/L yield was obtained with a tenfold dilution of the content of the dialysis bag with buffer

**Table 2. Parameters tested for the optimization of fast dilution refolding of histag-scFab fragments.**

| Refolding by dilution | | |
| --- | --- | --- |
| Parameters | Tested Conditions | Selected Conditions |
| L-Arginine | 0, 0.5, 1.0, 2.0 M | 2.0 M |
| Additives | 0.5–1.0 M Glycine<br>0.5% Tween-20<br>2.5 mM DTT | 2.5 mM DTT |
| Refolding temperature | 6, 21, and 37°C | 6°C |
| Refolding time | 48–96 hours | 60–65 hours |
| Sodium chloride | 0 and 150 mM | 0 mM |
| Protein concentration during refolding | 50–100 mg/L | 100 mg/L |

for affinity purification thereby reducing the advantage of sought with this approach. This suggests that a low concentration of arginine and larger sample volumes at this stage delay extensive precipitation. Attempts to remove oxidized glutathione led to poor yields, but substitution of reduced glutathione with dithiothreitol produced no change of the yield.

### Refolding parameter optimization

Amongst all parameters tested, pH had the most drastic effect on refolding yields. We tested pH between 5–11, and found pH 9 to be optimal, which appeared to be slightly above the calculated pI of the fragments (pI less than 9.0). However, isoelectric point measurements by cIEF were above 9.2 (Table 4). This result is somewhat counterintuitive as it indicates that refolding should be carried out at or above the fragment's pI, not below by more than 1 pH unit. This may explain why the low to no yields obtained with the on-column refolding method may be due, in part, to the lower pH (pH 8) utilized for the nickel column. The next important parameter was the L-arginine concentration. Higher yields were obtained with low concentration (0.5M) of L-arginine for the histag-NISTmAb-scFab while histag-trastuzumab-scFab refolded better at high concentration (2M). Keeping the refolding solution under reducing conditions with 2.5 mM DTT also improved refolding yields.

### Proteolytic cleavage of the polyhistidine tag and linker

All constructs have been engineered to allow removal of the polyhistidine fusion tag with thrombin protease and the poly-(GGGGS) linker with thrombin at the N-terminal end of the light chain and papain at the C-terminal end of the heavy chain. The initial strategy was to incubate with thrombin first, to remove the tag and nick the linker, and then to release the linker from the fragment with papain. While the tag is readily removed with 100 units of thrombin per milligram of protein, no cleavage is observed on the linker. Very large thrombin:fragment ratios were needed to only achieve about 70% of linker cleavage. This suggested that the thrombin recognition site (LVPRGS) is not easily accessible, therefore papain cleavage was then tested as a first step. Surprisingly, papain removed both the tag and the linker thus producing the targeted Fabs. Mass spectrometry analysis (Table 3) of all four fragments revealed that papain removes the polyhistidine tag by cutting after the thrombin recognition site, i.e., after Gly-Ser leaving no extra residue at the N-terminal end of the heavy chain, while cutting before the leucine of the thrombin recognition site thus leaving the site intact at the N-terminal end of the light chain. However, these bonus cuts came at a cost. Once complete cleavage of the linker was achieved, around 70–90% of the sample was degraded giving a 15 kDa broad and diffuse band on SDS-PAGE with the remainder being the Fab (S4 Fig). Various sources of papain (free and immobilized) and cleavage conditions (papain-to-fragment ratio, temperature, and incubation time) were evaluated to obtain these yields, which varied from no Fab to 10–30% cleaved product. In contrast, preparation of Fab fragments from papain cleavage of the corresponding therapeutic mAbs (Humira®, Rituxan® and Herceptin®) from the

**Table 3. Mass spectrometry analysis of papain cleaved histag-scFab from *E. coli* and corresponding Fab derived from commercialy acquired therapeutic mAb.**

|  | *E. coli* | | | Reference Drug | | |
|---|---|---|---|---|---|---|
|  | Expected | Measured | Difference | Expected | Measured | Difference |
| Adalimumab-Fab | 48,535.54 | 49,007.37 | 471.83 | 47,673.28 | 47,681.01 | 7.73 |
| NISTmAb-Fab | 48,485.80 | 48,960.02 | 474.22 | 47,636.54 | 47,630.77 | 5.77 |
| Rituximab-Fab | 48,050.99 | 48,522.17 | 471.18 | 47,204.73 | 47,178.31 | 26.42 |
| Trastuzumab-Fab | 48,487.48 | 48,960.49 | 473.01 | 47,629.20 | 47,636.96 | 7.76 |

**Table 4.  Isoelectric point determination and thermal unfolding monitored by circular dichroism.**

| | pI | | | $T_m$ (215 nm) * | |
|---|---|---|---|---|---|
| | Theoretical** | Reference*** | E. coli | Reference | E. coli |
| Adalimumab-Fab | 8.54 | 9.23 | 9.23 | 76.1 / 76.7 | 74.6 / 75.2 |
| NISTmAb-Fab | 8.65 | 9.68 | 9.81 | N/A | 74.5 / 73.7 |
| Rituximab-Fab | 8.88 | 9.61 | 9.65 | 77.3 / 77.0 | 75.1 / 75.2 |
| Trastuzumab-Fab | 8.64 | 9.35 | 9.55 | 83.2 / 82.1 | 81.8 / 80.1 |

* Two independent measurements

** Calculated using protparam website (https://web.expasy.org/protparam/)

*** Reference compounds are Humira®-Fab for adalimumab-Fab, NISTmAb-RM8761, Rituxan®-Fab for rituximab-Fab, and Herceptin®-Fab for trastuzumab-Fab.

marketplace produced pure Fab fragment without evidence of the low molecular weight impurities, or loss of the Fab fragment (S4 Fig).

## Assessment of the three-dimensional structure of Fab fragments by NMR spectroscopy

The methods of choice to assess the proper folding of these refolded Fab fragment are 2D-NMR proton-nitrogen and proton-carbon correlation maps such as $^1$H-$^{15}$N-SO-FAST-HMQC or $^1$H-$^{15}$N-HSQC and $^1$H-$^{13}$C-HSQC provided reference spectra are available [4]. In all cases, it is expected to observe extra resonances belonging to leftover residues of the thrombin recognition sequence (GS) on the amino terminal end of the light chain. The presence of these extra residues was also expected to induce small chemical shift perturbations (CSP) similar as those observed on spectra of the yeast prepared NISTmAb-Fab (S1 Fig) [7]. Also, reference spectra of all four Fab derived from therapeutic mAbs were all recorded at a slightly higher pH of 5.77, which will induce small CSP in solvent exposed residues. The overlay of the 2D-$^1$H-$^{15}$N-HSQC of $^{15}$N-labelled adalimumab-Fab prepared from E. coli and Humira®-Fab (natural abundance) is shown in Fig 2A. As expected, all backbone resonances of the E. coli prepared adalimumab-Fab are very well matched with reference drug Fab fragment, which confirm that the refolding process produced a properly folded conformation. Overlay of the 2D-$^1$H-$^{13}$C-HSQC of trastuzumab-Fab and Herceptin®-Fab is shown in Fig 2B. Spectral overlays of rituximab-Fab and Rituxan®-Fab and NISTmAb-Fab and the NISTmAb RM8761 (Figs 3 and S5) also reveal that the E. coli prepared Fabs show good matches with their reference drug-Fab spectra.

## Biophysical characterization: Isoelectric point, size exclusion chromatography, and melting temperature monitored by CD

Isoelectric points of Fabs derived from the parent monoclonal antibody and the corresponding E. coli prepared fragments shared very similar values (Table 4) that are systematically higher by 0.7–1.0 pH unit from the calculated values, which are known to be underestimated (web.expasy.org/compute_pi/pi_tool-doc.html). Similarly, melting temperatures monitored by circular dichroism of Fabs also showed similar values between E. coli prepared fragments and their reference fragment indicating that all Fabs are properly folded with same disulfide bonds topologies. In addition, the measured melting temperatures indicated that folds are very stable, such that NMR data collection can be carried out at temperatures as high as 50°C. Since fragments produced by all three refolding methods showed a great deal of protein precipitations during preparations, analysis of the oligomeric state of samples were carried out using size

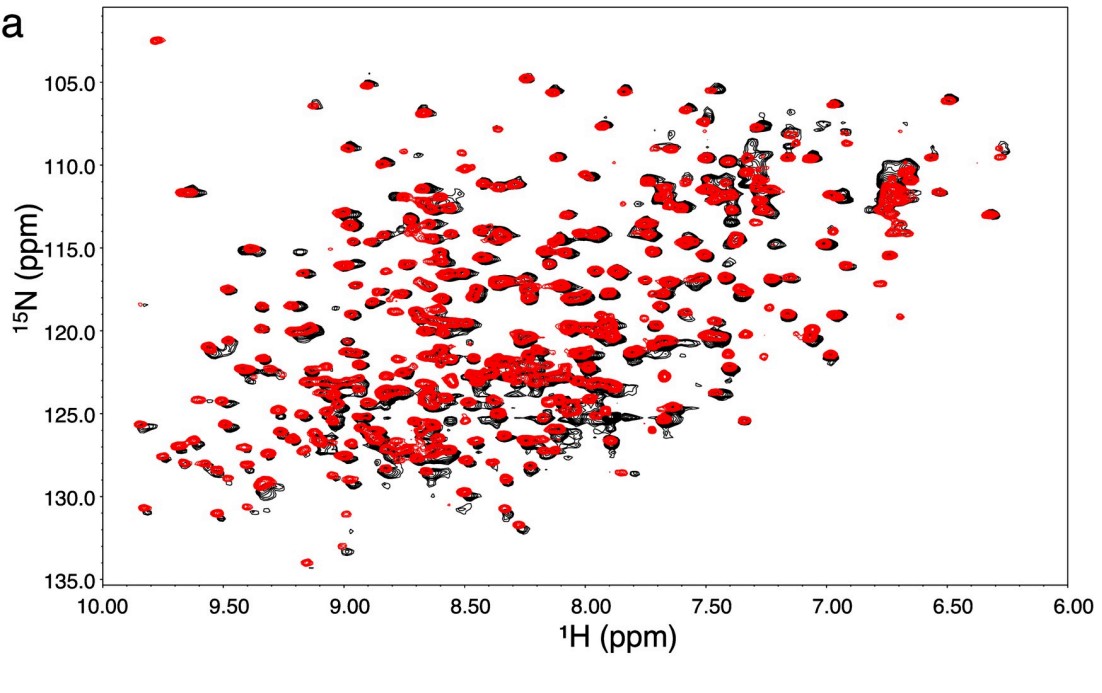

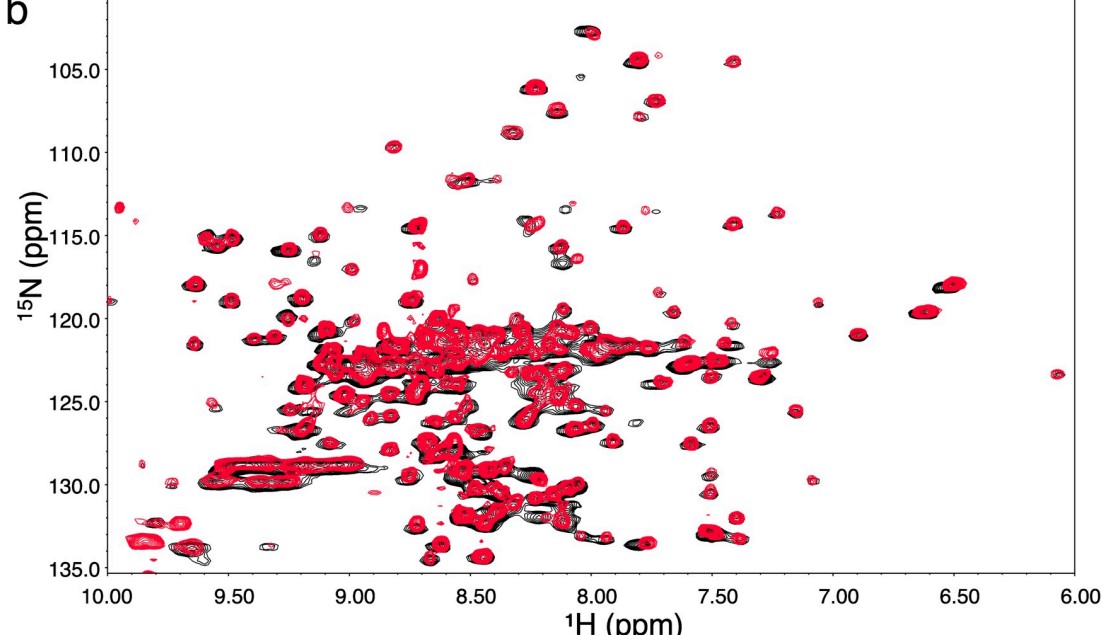

**Fig 2. NMR characterization of *E. coli*-produced Fabs.** A) Overlay of 2D-$^1$H-$^{15}$N-HSQC of $^{15}$N-labelled adalimumab-Fab prepared from *E. coli* (red) and Humira®-Fab (natural abundance) (black). B) Overlay of the methyl region of the 2D-$^1$H-$^{13}$C-HSQC of $^{13}$C-$^{15}$N -labelled trastuzumab-Fab prepared from *E. coli* (red) and Herceptin®-Fab (natural abundance) (black).

exclusion chromatography to detect the presence or absence of dimers and oligomers. Elution profiles are consistent with a single monomeric species, with very minor higher molecular weight species in two cases (S6 Fig).

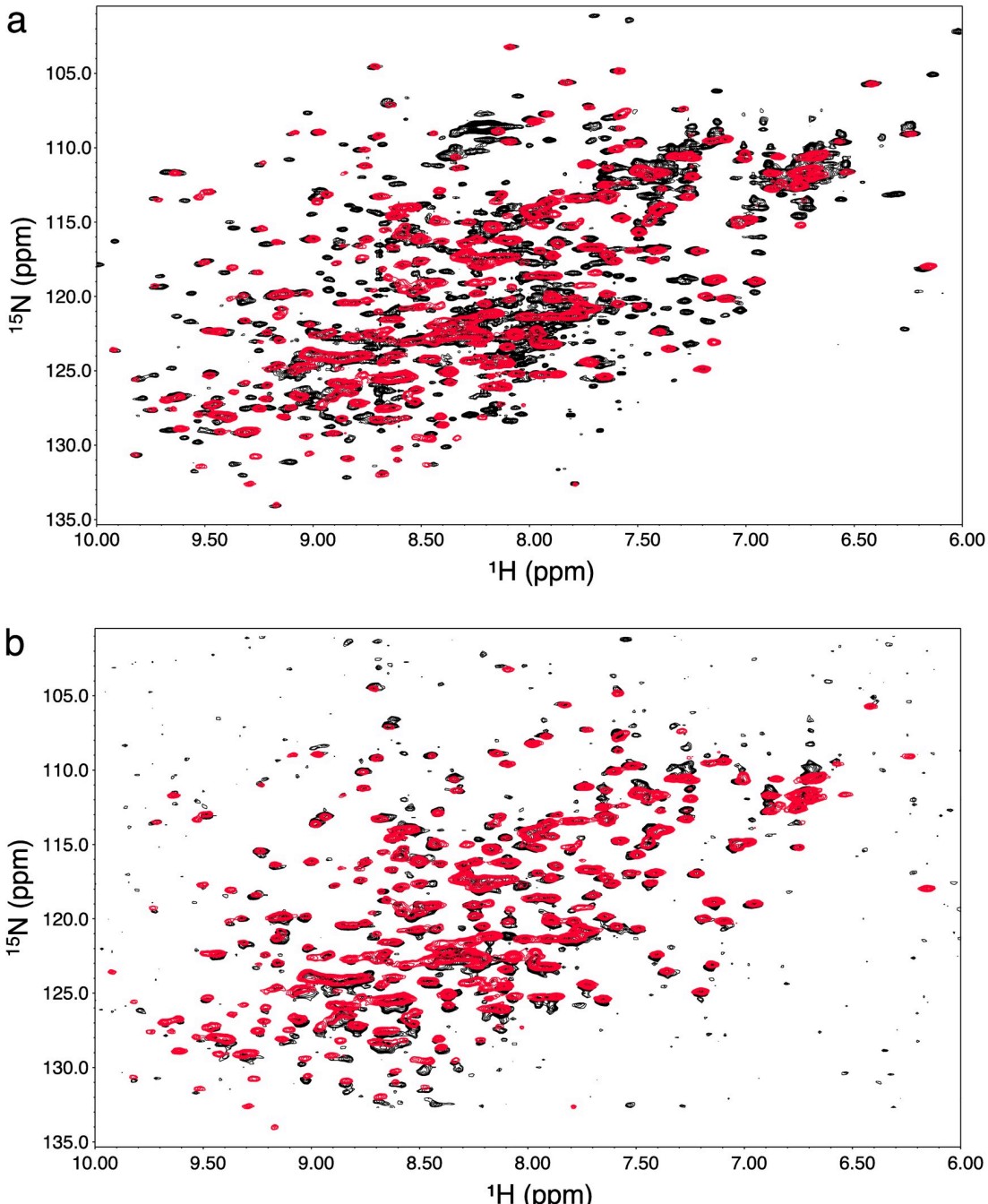

**Fig 3. Rituximab purified by IMAC and affinity chromatography.** A) Overlay of 2D-$^1$H-$^{15}$N-HSQC of $^{15}$N-labelled histag-rituximab-scFab purified from IMAC after fast dilution refolding over of $^{15}$N-labelled rituximab-Fab purified with CaptureSelect™ after fast dilution refolding with his tag and linker cleaved. B) Overlay of 2D-$^1$H-$^{15}$N-HSQC of $^{15}$N-labelled rituximab-Fab purified with CaptureSelect™ after fast dilution refolding with his tag and linker cleaved, and Rituxan®-Fab (natural abundance).

## Discussion

Production of Fab fragments for biophysical studies using NMR spectroscopy has been very challenging due to the requirement of isotopic enrichment. While incorporation of $^{13}$C and

[15]N isotopes can be achieved with various methods, complete or near complete substitution of protons ([1]H) with deuterons ([2]H) can only be achieved with yeast or bacteria. Our initial attempts were aimed at using *Komagataella phaffii* (*Pichia pastoris*) for its ability to express multi-domains with disulfide bond stabilized eukaryotic proteins with high levels of deuteration. In our hands, expression of the NISTmAb-Fab resulted in a well-folded fragment with an identical conformation to the Fab fragment derived from the NISTmAb RM 8671. However, preparation of a [2]H-[15]N-NISTmAb-Fab sample resulted in a significantly lower yield due to the difficulty of yeast cells to grow to high cell densities. For this reason, Brinson and co-workers elected to use a triply labelled rich growth media to attenuate this problem in order to improve protein yield in the production of [2]H-[13]C-[15]N-NISTmAb-Fab for resonance assignment [34]. In contrast, expression of a single chain version of Fab fragments in the form of inclusion bodies in *E. coli* was much simpler and more efficient. Refolding this abundant product, composed of beta-sheet elements and a total of five disulfide bridges, was where the challenge lays, especially when compared to smaller four-helix bundle proteins such as cytokines and hormones. Already, we observed that many refolding cycles were needed for the histag-NISTmAb-scFv, which is half the size of a Fab fragment. Fab fragments are composed of two chains, each containing two immunoglobulin domains, the latter being two beta-sheet elements stabilized by a disulfide bridge, and where the two chains are cross-linked via a fifth disulfide bridge at the carboxy-terminal. While the initial attempts at refolding the histag-NISTmAb-scFab that was immobilized on a nickel affinity column led to promising results, a plateau was reached where yields of properly folded fragments were low or null. Also, this method required many refolding cycles that consumed large amounts of guanidinium chloride. Instead of optimizing conditions for the on-column refolding method where only one set can be tested at a time per chromatographic system, it was more efficient to use the fast dilution method that allowed parallel parameter optimization. Amongst the various conditions tested (Table 2), pH 9 and arginine content (2 M) where the two changes that produced drastic increases in yields of properly folded fragments. The optimization process was also useful in testing reducing agents, namely reduced glutathione (GSH) and dithiothreitol (DTT), and testing whether oxidized glutathione was really needed for these fragments. One must maintain a ratio of GSH:GSSG of 1:4, however, a low concentration of DTT (2.5 mM) ensures that little or no oxygen is present in the refolding buffer at the start of the refolding. In view of the yields and practicality of the current conditions, refolding time (96 h) was optimized. Two reports [16, 17] on the refolding of Fab fragments have used refolding time of 120 and 150 hours. In our hands, similar results were obtained using 65 and 90 h refolding time.

A disadvantage of the fast dilution is the need to reduce the high concentration of arginine in the large solution volumes in order to allow the extraction of refolded proteins via affinity chromatography. Typically, the amount of inclusion bodies from a one-liter *E. coli* culture was enough such that two liters of refolding buffer was needed, thus producing equally large volume of sample to load on the first affinity purification step. For this reason, we tested the dialysis slow dilution refolding method. Here, the main advantage was the smaller volume of the refolding solution. However, this method often leads to lower yields resulting from higher levels of protein aggregation, that is anyhow protein dependent [35]. In our test, inclusion bodies produced from a one-liter *E. coli* culture were dissolved in 200–250 mL of denaturing buffer prior to serial dilutions. In this method, we only used 1 M arginine during the dilution steps, because of arginine solubility limitation in 6M GdmHCl. In the subsequent dilution steps, when the concentration of GdmHCl is 3 M or less, higher concentrations of arginine could be used thereby offering the same flexibility of conditions optimization. However, injection of the refolded solution on the affinity resin led to lower yields compared to tenfold dilution prior to affinity chromatography purification. This suggested that the refolding volume may be too

small (too high concentration of inclusion bodies), which would allow more unwanted inter-molecular interactions leading to aggregation, and that low amounts of arginine may prevent precipitation of properly folded material along with improperly or partially folded fragment. Considering the pros and cons of both refolding methods, the fast dilution method is, in our hands, a technically simpler and faster method to refold scFab fragments.

Isolation of properly refolded Fab fragments was best accomplished using affinity chroma-tography with CaptureSelect™ resin that binds the $C_H1$ domain of the fragment. Initially, low to moderate recovery yields were obtained when eluted with glycine buffer at pH 3.0. Yields were significantly improved by either lowering the pH to 2.5 or using acetic acid at pH 2.5. Higher yields were obtained by performing two elutions of the resin. Although the polyhisti-dine tag was fused to the scFab fragments with the intent to carry out the on-column refolding method, we tested nickel-affinity purification as an alternative method to purify the target frag-ments. The results showed that properly folded, as well as misfolded fragments co-purify using IMAC (Fig 3A) while the CaptureSelect™ resin selectively binds to properly folded histag-scFab fragments (Fig 3B).

Removal of the polyhistidine tag and linker could be achieved in one step, albeit with a sig-nificant loss of the refolded fragments. Initially, NMR analysis of the refolded fragments indi-cated the presence of misfolded species that co-eluted with properly folded fragments. Therefore, papain cleavage presumably cleaned up refolded preparation by removing those mis-folded species and provided samples of properly folded fragments only. Extra manipula-tions of the sample following affinity chromatography using CaptureSelect™, such as additional buffer exchanges, yielded essentially pure well-folded Fab fragments. Note that these additional exchanges were also performed following IMAC yet improperly folded species were not totally eliminated. However, papain treatment destroyed a significant portion of the well folded frag-ment (70–90%), which indicated that other enzymes need to be considered for removing the linker. Work is underway to replace both proteases with tobacco etch virus protease. A priori, the presence of the linker can be tolerated, since scFv fragments are known to retain most, if not all, of their binding affinities with their target. Nevertheless, we successfully prepared fully cleaved fragments for characterization and compared them with the corresponding Fab derived from the parent drug.

## Characterization of refolded Fab fragments

Assessment of the bioactive conformation was carried out by comparing 2D-NMR spectra of Fabs from the parent drugs and *E. coli*-prepared material. Bioassays were not considered since the demonstration of virtually identical NMR fingerprints between refolded Fab and reference Fab is a far superior assessment of conformation than bioactivity. A partly misfolded fragment could very well bind its target via its CDR elements and conserve activity while having other regions of the fragment being improperly folded. We have shown that single point mutants of filgrastim can have no effect on the activity in bioassay but still exhibited many very distinctive chemical shift perturbations [32].

All reference two-dimensional proton-nitrogen spectra were recorded with the $^1$H-$^{15}$N-SO-FAST-HMQC pulse sequence that produced broader resonances resulting from a lower digital resolution, while all Fab spectra prepared from *E. coli* were recorded with the HSQC sequence that provides twice the resolution of the SOFAST version in the proton dimension. RF power limitation of cryoprobe limits the acquisition time to 50 milliseconds, compared to 100 ms used on conventional HSQC pulse sequences, thereby reducing digital resolution by two-fold. In addition, all Fab of marketed drugs were dissolved in deuterated sodium acetate at pH 5.77, which induced small chemical shift differences in exposed loops. In contrast, overlay of

$^1$H-$^{13}$C-HSQC spectra produced better resonance matches since these correlations are less sensitive to variations of the local magnetic environments. However, the lack of resolution in the crowded region ($^1$H 0.5–1.3ppm and $^{13}$C 20-23ppm) somewhat limits detailed resonance assessment. We did not attempt to use statistical analysis tools such as principal component analysis [36] to push the analysis further. The visual analysis of the overlay was deemed sufficient for our purpose.

## Conclusion

The strategy described here takes advantage of the facile construction of bacterial expression constructs, and can provide multi-milligram amounts of Fab fragments for high-resolution NMR studies and other approaches. Correct folding of the end product was readily assessed by comparing it with 2D-NMR spectra of the target drug recorded at natural abundance. Production of triply labelled adalimumab and trastuzumab with high isotopic incorporation (99% $^2$H, $^{13}$C, $^{15}$N) has been carried out with protocols described here. Samples suitable for NMR resonance assignments have been prepared. In view of the lower protein yields from our work with *K. phaffii* and considering the more involved construction of the expression system, *E. coli* thus provides a more economical approach to produce isotopically labelled samples, especially fully deuterated samples.

## Supporting information

**S1 Fig. Initial results from the production of yeast $^{15}$N-NIST-mAb-Fab.** A) Overlay of 2D-$^1$H-$^{15}$N-SOFAST-HMQC of $^{15}$N-labelled isolate from yeast $^{15}$N-NIST-mAb-Fab isolated using cation exchange chromatography (blue) over NIST-mAb-Fab RM-8761 at natural abundance (black). The spectrum in blue shows high intensity peaks with distorted lineshape that are attributed to the light chain while peaks of lower intensities belong to the Fab. B) Overlay of yeast-produced $^{15}$N-NIST-mAb-Fab (red) and RM-8761 at natural abundance (black) showing the extra resonances arising from the tetrapeptides EAEA at the N-terminal end of the heavy and light chains.
(JPG)

**S2 Fig. SDS-PAGE analysis of protein expression.** SDS-PAGE analysis of protein induction times (in hours indicated at the top of every gel) at two temperatures for all four Fab fragments. The target protein (histag-mAb-scFab c.a. 55 kDa) is boxed in red.
(JPG)

**S3 Fig. Initial result of refolding $^{15}$N-histag-NISTmAb-scFab.** Overlay of 2D-$^1$H-$^{15}$N-SOFAST-HMQC of $^{15}$N-yeast-NISTmAb-Fab (red) and first very diluted sample of $^{15}$N-histag-NISTmAb-scFab prepared with IMAC on-column refolding from *E. coli* inclusion bodies (black).
(JPG)

**S4 Fig. Biophysical characterization of the various Fab.** A) SDS-PAGE analysis of papain cleavage of histag-mAbs-scFabs. Ratios of scFab:Papain tested were 25:1;50:1, 100:1 and 250:1. B) SDS-PAGE analysis of papain cleavage of innovator therapeutic mAbs-Fabs. Letters U and C correspond to uncleaved and cleaved, respectively. C) Mass spectrometry analysis revealed an extra 472 mass units of the papain cleavage product on the non-reduced samples. Analysis of the reduced fragments shows that the extra residues are at the amino-terminal of the light chain. D) Amino acid sequence at the end of the linker and beginning of the light chain showing the mass of every residue underneath.
(JPG)

**S5 Fig. Comparison of *E.coli* prepared $^{15}$N-NISTmAb-Fab with the reference material.**
Overlay of 2D-$^1$H-$^{15}$N-HSQC of $^{15}$N-NISTmAb-Fab (tag and linker cleaved) (red) and
2D-$^1$H-$^{15}$N-SOFAST-HMQC of NIST-mAb-Fab RM-8761 at natural abundance (black).
(JPG)

**S6 Fig. S6 size-exclusion profiles for the four Fab fragments.** A) SEC analysis of *E.coli* pre-
pared trastuzumab-Fab prior to cleavage of the histag and linker (histag-trastuzumab-scFab)
and after cleavage (trastuzumab-Fab) freshly cleaved and after spending several hours in the
NMR spectrometer at 50°C. Note the presence of an unknown impurity at ∼17.5min of lower
molecular weight (denoted by a red star) that nearly disappeared after spending time at 50°C.
B) Comparison of SEC analysis of *E.coli* prepared Fab and the Fab from the corresponding
drug. In all preparations, but this preparation of trastuzumab, the impurity (red star) is pres-
ent.
(JPG)

**S1 Table. Composition of the different vectors.**
(DOCX)

**S2 Table. Sequences of the different primers.**
(DOCX)

**S1 Raw images.**
(PDF)

**S2 Raw images.**
(PDF)

# Acknowledgments

The authors thank Dr. Roger Tam and Dr. Simon Sauvé for critical reading of the manuscript
and Dr. Robert Brinson and Dr. John Marino for the donation of the Fab domain from the
NISTmAb-Fab RM8761 reference material.

# Author Contributions

**Conceptualization:** Muzaddid Sarker, Yves Aubin.

**Formal analysis:** Donald Gagné, Muzaddid Sarker, Geneviève Gingras, Yves Aubin.

**Investigation:** Donald Gagné, Muzaddid Sarker, Geneviève Gingras, Yves Aubin.

**Methodology:** Donald Gagné, Muzaddid Sarker, Geneviève Gingras, Derek J. Hodgson,
Grant Frahm, Marybeth Creskey, Barry Lorbetskie, Stewart Bigelow, Jun Wang, Xu Zhang,
Michael J. W. Johnston, Huixin Lu, Yves Aubin.

**Project administration:** Yves Aubin.

**Resources:** Yves Aubin.

**Supervision:** Yves Aubin.

**Visualization:** Muzaddid Sarker, Geneviève Gingras, Derek J. Hodgson, Grant Frahm,
Marybeth Creskey, Barry Lorbetskie, Stewart Bigelow, Jun Wang, Xu Zhang,
Michael J. W. Johnston, Huixin Lu.

**Writing – original draft:** Donald Gagné, Yves Aubin.

**Writing – review & editing:** Donald Gagné, Muzaddid Sarker, Xu Zhang, Michael J. W. Johnston, Huixin Lu, Yves Aubin.

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
