## [Decision Letter · Decision Letter 0]

6 Oct 2023

PONE-D-23-28318Strategies for the production of isotopically labelled Fab fragments of therapeutic antibodies in Komagataella phaffii (Pichia pastoris) and Escherichia coli for NMR studiesPLOS ONE

Dear Dr. Aubin,

Thank you for submitting your manuscript to PLOS ONE. After careful consideration, we feel that it has merit but does not fully meet PLOS ONE’s publication criteria as it currently stands. Therefore, we invite you to submit a revised version of the manuscript that addresses the points raised during the review process.

We look forward to receiving your revised manuscript.

Kind regards,

Rui Tada, Ph.D.

Academic Editor

PLOS ONE

Journal Requirements:

3. Thank you for stating the following financial disclosure: "Funded by internal funds from the Government of Canada, Health Department"

Reviewers' comments:

Reviewer's Responses to Questions

**Comments to the Author**

1. Is the manuscript technically sound, and do the data support the conclusions?

Reviewer #1: Yes

Reviewer #2: Yes

2. Has the statistical analysis been performed appropriately and rigorously? 

Reviewer #1: N/A

Reviewer #2: N/A

3. Have the authors made all data underlying the findings in their manuscript fully available?

Reviewer #1: Yes

Reviewer #2: Yes

4. Is the manuscript presented in an intelligible fashion and written in standard English?

Reviewer #1: Yes

Reviewer #2: Yes

5. Review Comments to the Author

Reviewer #1: In the present manuscript, the authors describe efficient production of Fabs in Komagatella phaffii and E. coli and preparation of the material for 2D-1H-15N-HSQC analysis. Both systems were functional in Fab production, but refolding from E. coli-produced inclusion bodies was finally chosen (mostly because of lower cost), with yields of several ten milligrams per liter of initial culture. For refolding, they choose fast dilution method, and for functional Fab they used affinity purification over Capture Select CH1 column, as Ni-NTA method resulted in elution of a mix of folded and unfolded molecules. Apart from NMR analysis of isotopically labelled Fabs, the authors also determined their SEC profiles, their thermostability using thermo-CD, their isoelectric point, and analysed them with mass spectrometry. Importantly, they compared their material with Fab fragments produced with cleavage of commercially acquired therapeutic antibodies. Altogether, this is an important contribution to development of protocols supportive to antibody fragment material preparation for NMR analysis, which is gaining in interest as a method of comparison of biosimilars.

The “Materials and methods” part of the manuscript is extremely well written, including very detailed descriptions of all procedures and the optimization steps, which I am sure the audience will appreciate. Also, the Results are well presented, the inference is clear and in spite of large amounts of data, the text is interesting to read. Figure legends (especially to Supplementary Figures) could benefit from improvement and more details should be included (please see remarks below). I would also propose a careful re-read of the Discussion section by all authors (there are words missing and sentences sometimes do not make sense).

Please find below a list of remarks which I hope will be helpful.

Page 2: Abstract: word order: “of immunoglobulin G1-based therapeutic antibodies“ - of therapeutic antibodies of immunoglobulin G1 kappa isotype.

Page 3: Reference 1 is not so suitable – the review “Antibodies to watch” is written every year and the 2022 issue should be cited.

Page 5: “In addition, the yeast K. phaffii has the ability to produce N-glycosylated proteins in Fc fragments of mAb“: this should probably be: …has the ability to produce N-glycosylated Fc fragments. It should also be briefly mentioned that K.phaffii glycosylation is different from human.

Page 16: is this a HiTrap SP FF prepacked column?

Page 17:All NMR spectra were processed

Page 20: Fab peaks

Page 22: In the Figure S2, only results for 16°C and 37°C-induction are presented

Page 24: adalimumab-scFab showed higher yield of refolded protein – please specify comparing to what?

Page 24: set out to optimize refolding conditions“

Table 2: the column titles should be “tested conditions” and “selected conditions”

Page 25: “were prepared for the final production of Fab fragments“ is not a great sentence because the protocol described here is already a part of production.

Page 27: pure Fab fragment is better than “cleaned Fab fragment”, and loss of the Fab fragment (of is missing)

Page 27: Title to Table 3: “from marketplace derived“ is not a good expression and word order is not OK. Maybe: and corresponding Fab derived from commercially acquired therapeutic mAb

Legend to Figure S4: please add more information, what are the numbers in the lane labels in A, and what is U and C in B, uncleaved and cleaved?

Figure S6: Please add gel filtration standard and indicate the sizes of marker proteins.

Page 28: CSP (Chemical shift perturbation) abbreviation not explained

Page 29: in SC, all antibody-derived Fabs except trastuzumab have a second minor peak of lower molecular weight – what is that?

Table S1: please use uniform capitalization in the vector name (pPICZalphaA) and restriction enzyme (PmeI). Bicistron should be bicistronic.

Page 30: (references) – did you intend to include more references?

Page 31: challenge

Discussion Page 31: is the size of the protein that makes the refolding difficult, or the structural features such as number of disulphide bridges etc?

Page 32: “In our hands 65 and 90 hours did not show observable differences“ – this sentence is too colloquial, maybe: the refolding time between 65 and 90 h resulted in similar results – it would also be good to specify what the results were (product quality?)

Page 32: “a one-liter E. coli culture generate enough inclusion bodies“ – the amount of inclusion bodies produced from 1-L-E.coli culture, or similar

Page 32: “This approach used only 1M arginine during the dilution step because higher

concentrations of arginine, such as 2M used in the fast dilution method, due to solubility

limitation in the presence of 6M GdmHCl“, something is wrong with this sentence.

Page 33: CH1, 1 is not in subscript

Page 33: papain treatment destroyed a significant portion of the well folded fragment, of is missing

Page 34: “for characterization them and compare them“: should be for characterization and compared them

Page 35: Conclusion: last sentence is a bit out of place, as there are no data shown. Regarding that you show you have 4 good Fabs, including fully human, humanized and chimeric, proves already that the method is applicable to many different ones.

Reviewer #2: In their manuscript, Gagné et al. describe different expression strategies for isotopically labeled Fab fragments of antibodies. Experiments are conducted with both E. coli and yeast. Using a variety of complementary approaches, they assess the efficiency of different expression and purification protocols, focusing on the refolding steps after purification from inclusion bodies. They use NMR spectroscopy as a tool to verify the foldedness of the antibodies obtained in this manner in comparison to commercially available antibody samples (at natural abundance).

This report represents a significant amount of experimental work that appears to have been carried out meticulously. The paper is well-organized and reasonably well-written (however, see below), and the results are interesting for the addressed scientific community. I support its publication in PLOS ONE Resonance. However, several issues should be resolved before acceptance:

1. On page 23, the procedure of repeated cycles of refolding is confusing. The authors state: "The denaturant is removed via a gradient (linear or else) with the same buffer without denaturant under oxidative conditions prior to elution of the protein. The procedure can be repeated by re- equilibrating the column with the denaturing buffer, then the cycle of denaturant removal via a gradient and protein elution is repeated to obtain additional amounts of folded protein." This suggests that only folded protein is eluted from the affinity resin, which does not make sense. Please elaborate on this issue.

2. The sentence starting with "In our test, inclusion bodies..." makes no sense.

3. Caption Figure 2: Color coding is not clear (which one is the red spectrum, and which one is the black spectrum?). Both in A) and B). Also in Figure S1: Color coding is unclear; as a matter of fact the entire first half of the figure caption is unclear...

4. A technical note: Page 13: "A cycle starts with a linear gradient from 100% to 50% of buffer G against buffer B (...) at 2 mL/min...". The length of the procedure is missing (column volumes or mL).

5. On pages 13/14, drop-by-drop dilution refolding: Does "The cell pellet ..." refer to the pellet that was obtaind from 1 L of growth medium? Was the resuspension volume normalized by measuring the cell density of the expression culture before harvest? For the refolding process itself the refolding buffer is described properly, but its volume is missing.

6. Page 26: "Isoelectric point measurements by cIEF were above 9.2 while the calculated values were less than 9.0 (Table 4) indicating that refolding should be carried out at or above the fragment’s pI, not below by more than 1 pH unit." I am having difficulty comprehending the argument. Why not use a pH for refolding that is well below the experimentally determined pI? And why is it suggested to use a pH that is "at the fragment's pI"? I would have assumed that staying away from the pI is ideal?

7. Please comment on how the weight of the inclusion bodies was determined. By weighing pellets? On page 22, bottom, the statement "all fragments produced large amounts of inclusion bodies (c.a. 114-140 mg per liter of culture measured in 6 M guanidinium chloride)" suggests that pellet weight was determined in 6 M guanidinium chloride. How was this done?

8. More technical notes: On page 10, buffer A is the only one that is given in g/L rather than mM. And why was the use of deuterated Tris buffer necessary?

9. "(references)". Are references missing?

10. On page 34, the authors state: "Bioassays were not considered since the demonstration of virtually identical NMR fingerprints between refolded Fab and reference Fab is a far superior assessment of conformation than bioactivity." and "We have shown that single point mutants of filgrastim can have no effect on the activity in bioassay but still exhibited many very distinctive chemical shift perturbations". There seems to be a contradiction. The first sentence argues for using NMR for assessment as opposed to bioassays, while the second sentence would suggest bioassays to be superior.

11. Page 34, last paragraph: "wider resonances" is not a commonly used term. Broader signals is.

12. Typos: Page 4 bottom: if Frab fragments; pages. Page 5 center: polypeptide chain that are; Page 6 center: all exchangeable backbone amide fully protonated. Page 17 center: spectra were process with nmrPipe; same paragraph: transferred to 5 mm tube. Page 22 top: Attempt to produce. Page 29, bottom: indicating that fold are very stable. Page 30: achieved with the yeast. Page 31 top. a triply labelled rich growth media. Same page: where the challenged laid. Page 32: a ... culture generate enough. Page 34 first paragraph: characterization them.

The manuscript would certainly benefit from careful proofreading, in particular in the Results and Discussion sessions.

6. PLOS authors have the option to publish the peer review history of their article (what does this mean?). If published, this will include your full peer review and any attached files.

Reviewer #1: No

Reviewer #2: No

---

## [Author Response · Author response to Decision Letter 0]

25 Oct 2023

Response to reviewers

We thank reviewers 1 and 2 for their in-dept review and constructive comments of our manuscript. They both have pointed out a number of imprecisions and minor errors that we were glad to correct. Overall, these reviews have clearly helped us improving the clarity and precision of the manuscript. We have addressed all points raised by the reviewers and corrected, or reworded the manuscript accordingly. In a couple of cases, we explained or answered questions raised by reviewers. We trust that our answers will address reviewers’ concerns.

Reviewer #1: In the present manuscript, the authors describe efficient production of Fabs in Komagatella phaffii and E. coli and preparation of the material for 2D-1H-15N-HSQC analysis. Both systems were functional in Fab production, but refolding from E. coli-produced inclusion bodies was finally chosen (mostly because of lower cost), with yields of several ten milligrams per liter of initial culture. For refolding, they choose fast dilution method, and for functional Fab they used affinity purification over Capture Select CH1 column, as Ni-NTA method resulted in elution of a mix of folded and unfolded molecules. Apart from NMR analysis of isotopically labelled Fabs, the authors also determined their SEC profiles, their thermostability using thermo-CD, their isoelectric point, and analysed them with mass spectrometry. Importantly, they compared their material with Fab fragments produced with cleavage of commercially acquired therapeutic antibodies. Altogether, this is an important contribution to development of protocols supportive to antibody fragment material preparation for NMR analysis, which is gaining in interest as a method of comparison of biosimilars.

The "Materials and methods" part of the manuscript is extremely well written, including very detailed descriptions of all procedures and the optimization steps, which I am sure the audience will appreciate. Also, the Results are well presented, the inference is clear and in spite of large amounts of data, the text is interesting to read. Figure legends (especially to Supplementary Figures) could benefit from improvement and more details should be included (please see remarks below). I would also propose a careful re-read of the Discussion section by all authors (there are words missing and sentences sometimes do not make sense).

Please find below a list of remarks which I hope will be helpful.

Page 2: Abstract: word order: "of immunoglobulin G1-based therapeutic antibodies" - of therapeutic antibodies of immunoglobulin G1 kappa isotype. 

Response: manuscript was corrected

Page 3: Reference 1 is not so suitable - the review "Antibodies to watch" is written every year and the 2022 issue should be cited.

Response: manuscript was corrected

Page 5: "In addition, the yeast K. phaffii has the ability to produce N-glycosylated proteins in Fc fragments of mAb": this should probably be: ...has the ability to produce N-glycosylated Fc fragments. It should also be briefly mentioned that K.phaffii glycosylation is different from human.

Response: this section manuscript was reworded to capture this fact

Page 16: is this a HiTrap SP FF prepacked column? 

Response: Yes, manuscript was corrected.

Page 17: All NMR spectra were processed

Response: manuscript was corrected

Page 20: Fab peaks

Response: manuscript was corrected

Page 22: In the Figure S2, only results for 16°C and 37°C-induction are presented

Response: Initially, we had carried expressions at 25°C but subsequently removed these from the paper because it brought nothing additional so manuscript was corrected

Page 24: adalimumab-scFab showed higher yield of refolded protein - please specify comparing to what?

Response: text was added: …compared to previous attempts, 

Page 24: set out to optimize refolding conditions" 

Response: manuscript was corrected

Table 2: the column titles should be "tested conditions" and "selected conditions"

Response: Table was modified

Page 25: "were prepared for the final production of Fab fragments" is not a great sentence because the protocol described here is already a part of production. 

Response: Sentence has been rewritten.

Page 27: pure Fab fragment is better than "cleaned Fab fragment", and loss of the Fab fragment (of is missing)

Response: Manuscript has been corrected

Page 27: Title to Table 3: "from marketplace derived" is not a good expression and word order is not OK. Maybe: and corresponding Fab derived from commercially acquired therapeutic mAb

Response: Sentence has been reworded.

Legend to Figure S4: please add more information, what are the numbers in the lane labels in A, and what is U and C in B, uncleaved and cleaved?

Response: Legend has been modified.

Figure S6: Please add gel filtration standard and indicate the sizes of marker proteins. 

Response: a new S6 figure and legend have been generated to address this point and the next 

Page 29: in SC, all antibody-derived Fabs except trastuzumab have a second minor peak of lower molecular weight - what is that?

Response: Short answer: we don’t know. However, we included an SEC analysis of the several trastuzumab samples. This analysis (panel A of Figure S6) shows this peak disappearing when the sample is subjected to 50oC for a few hours. The trace in panel B for trastuzumab may have been anayzed by NMR prior to do the SEC. 

Page 28: CSP (Chemical shift perturbation) abbreviation not explained

Response: The abbreviation has been added after the first mention of Chemical shift perturbation.

Table S1: please use uniform capitalization in the vector name (pPICZalphaA) and restriction enzyme (PmeI). Bicistron should be bicistronic.

Response: corrected

Page 30: (references) - did you intend to include more references? 

Response: No need for references here, this was a reminder that we overlooked.

Page 31: challenge

Discussion Page 31: is the size of the protein that makes the refolding difficult, or the structural features such as number of disulphide bridges etc?

Response: Yes, we added details to indicate this.

Page 32: "In our hands 65 and 90 hours did not show observable differences" - this sentence is too colloquial, maybe: the refolding time between 65 and 90 h resulted in similar results - it would also be good to specify what the results were (product quality?)

Response: Text has been reworded. Refolding time did not have any effects on yield or quality.

Page 32: "a one-liter E. coli culture generate enough inclusion bodies" - the amount of inclusion bodies produced from 1-L-E.coli culture, or similar

Response: This was corrected

Page 32: "This approach used only 1M arginine during the dilution step because higher

concentrations of arginine, such as 2M used in the fast dilution method, due to solubility. limitation in the presence of 6M GdmHCl", something is wrong with this sentence. 

Response, we shortened the sentence to clarify this point. 

Page 33: CH1, 1 is not in subscript

Response: corrected. We used the following notation XY¬#, where X= C or V; Y= H or L; #= 1,2,3.

Page 33: papain treatment destroyed a significant portion of the well folded fragment, of is missing

Response: corrected

Page 34: "for characterization them and compare them": should be for characterization and compared them

Response: extra word was removed

Page 35: Conclusion: last sentence is a bit out of place, as there are no data shown. Regarding that you show you have 4 good Fabs, including fully human, humanized and chimeric, proves already that the method is applicable to many different ones.

Response: Indeed this sentence was not needed and was removed.

Reviewer #2: In their manuscript, Gagné et al. describe different expression strategies for isotopically labeled Fab fragments of antibodies. Experiments are conducted with both E. coli and yeast. Using a variety of complementary approaches, they assess the efficiency of different expression and purification protocols, focusing on the refolding steps after purification from inclusion bodies. They use NMR spectroscopy as a tool to verify the foldedness of the antibodies obtained in this manner in comparison to commercially available antibody samples (at natural abundance).

This report represents a significant amount of experimental work that appears to have been carried out meticulously. The paper is well-organized and reasonably well-written (however, see below), and the results are interesting for the addressed scientific community. I support its publication in PLOS ONE Resonance. However, several issues should be resolved before acceptance:

1. On page 23, the procedure of repeated cycles of refolding is confusing. The authors state: "The denaturant is removed via a gradient (linear or else) with the same buffer without denaturant under oxidative conditions prior to elution of the protein. The procedure can be repeated by re- equilibrating the column with the denaturing buffer, then the cycle of denaturant removal via a gradient and protein elution is repeated to obtain additional amounts of folded protein." This suggests that only folded protein is eluted from the affinity resin, which does not make sense. Please elaborate on this issue.

Response: This method was first developed for the prion protein by the group of Wuthrich. In our vast experience (prion protein, GM-CSF and mutants, G-CSF and mutants, interleukin-2a and 2b and mutants, hGH, NISTmAb-scFv it was exactly that. Only refolded protein elutes off the column. Whatever refolded protein formed on the resin eluted off, while all other species were obviously not soluble and remained in the column. With 2-4 cycles, we often obtained nearly complete refolding of our inclusion bodies. However, Fab fragments can form soluble but improperly folded species that end up precipitating after elution. That was a first. 

2. The sentence starting with "In our test, inclusion bodies..." makes no sense. 

Response, (p.32) the word ‘corresponding’ has been changed with ‘produced from’ should clarify the point.

3. Caption Figure 2: Color coding is not clear (which one is the red spectrum, and which one is the black spectrum?). Both in A) and B). Also, in Figure S1: Color coding is unclear; as a matter of fact, the entire first half of the figure caption is unclear...

Response: Thank you very much for picking this up! We modified the legend to add the colours. In addition, the S1 legend was rewritten to clarify the point about the blue spectrum. 

4. A technical note: Page 13: "A cycle starts with a linear gradient from 100% to 50% of buffer G against buffer B (...) at 2 mL/min...". The length of the procedure is missing (column volumes or mL).

Response: the duration for each gradient was 10 CV, the text was updated.

5. On pages 13/14, drop-by-drop dilution refolding: Does "The cell pellet ..." refer to the pellet that was obtained from 1 L of growth medium? Was the resuspension volume normalized by measuring the cell density of the expression culture before harvest? For the refolding process itself the refolding buffer is described properly, but its volume is missing.

Response: Yes, IB from a one-liter culture refolded in a 2 L refolding buffer solution. The amount of IB was measured after resuspension in 20 mL GdmCl to accommodate all various IB yields, such that we never exceed the final 100 ug/L of refolded protein.

6. Page 26: "Isoelectric point measurements by cIEF were above 9.2 while the calculated values were less than 9.0 (Table 4) indicating that refolding should be carried out at or above the fragment's pI, not below by more than 1 pH unit." I am having difficulty comprehending the argument. Why not use a pH for refolding that is well below the experimentally determined pI? And why is it suggested to use a pH that is "at the fragment's pI"? I would have assumed that staying away from the pI is ideal?

Response: Initially, we assumed the same thing as reviewer 2. However, pHs of 5-11 (see text) were tested and only pH 8 afforded sufficient amounts of refolded Fab, while quantities were optimal at pH 9. Higher pHs did worked as well but with small decreases of yields so we elected to be a 9, which is at the pI. The sentence was reworded to clarify this.

7. Please comment on how the weight of the inclusion bodies was determined. By weighing pellets? On page 22, bottom, the statement "all fragments produced large amounts of inclusion bodies (c.a. 114-140 mg per liter of culture measured in 6 M guanidinium chloride)" suggests that pellet weight was determined in 6 M guanidinium chloride. How was this done?

Response: The title of Table 1 was modified to include ‘measured by absorbance in 6M GdmCl’. The cell pellets were simply weighted after centrifugation.

8. More technical notes: On page 10, buffer A is the only one that is given in g/L rather than mM. And why was the use of deuterated Tris buffer necessary?

Response: Gram per liter are easier to work with as one can see from the added concentrations in mM in bracket) Deuterated Tris was needed ‘for NMR analysis’. Added in the text 

9. "(references)". Are references missing?

Response: left from a previous version that escaped out proof reading. No reference needed there.

10. On page 34, the authors state: "Bioassays were not considered since the demonstration of virtually identical NMR fingerprints between refolded Fab and reference Fab is a far superior assessment of conformation than bioactivity." and "We have shown that single point mutants of filgrastim can have no effect on the activity in bioassay but still exhibited many very distinctive chemical shift perturbations". There seems to be a contradiction. The first sentence argues for using NMR for assessment as opposed to bioassays, while the second sentence would suggest bioassays to be superior.

Response: That’s the power of NMR: chemical shift perturbations will reveal the presence of a mutation while the bioassay can miss it by showing the same biological activity. 

11. Page 34, last paragraph: "wider resonances" is not a commonly used term. Broader signals is.

Response: yes indeed, corrected.

12. Typos: Page 4 bottom: if Frab fragments; pages. Page 5 center: polypeptide chain that are; Page 6 center: all exchangeable backbone amide fully protonated. Page 17 center: spectra were process with nmrPipe; same paragraph: transferred to 5 mm tube. Page 22 top: Attempt to produce. Page 29, bottom: indicating that fold are very stable. Page 30: achieved with the yeast. Page 31 top. a triply labelled rich growth media. Same page: where the challenged laid. Page 32: a ... culture generate enough. Page 34 first paragraph: characterization them.

Response: all were corrected except labelled rich growth media: it is a rich media, vs a minimal media that is labelled.

The manuscript would certainly benefit from careful proofreading, in particular in the Results and Discussion sessions.

Response: thank you for your careful reading. We have looked again at the manuscript to identify typos and errors.

---

## [Decision Letter · Decision Letter 1]

31 Oct 2023

Strategies for the production of isotopically labelled Fab fragments of therapeutic antibodies in Komagataella phaffii (Pichia pastoris) and Escherichia coli for NMR studies

PONE-D-23-28318R1

Dear Dr. Aubin,

We’re pleased to inform you that your manuscript has been judged scientifically suitable for publication and will be formally accepted for publication once it meets all outstanding technical requirements.

Kind regards,

Rui Tada, Ph.D.

Academic Editor

PLOS ONE

Reviewers' comments:

Reviewer's Responses to Questions

**Comments to the Author**

1. If the authors have adequately addressed your comments raised in a previous round of review and you feel that this manuscript is now acceptable for publication, you may indicate that here to bypass the “Comments to the Author” section, enter your conflict of interest statement in the “Confidential to Editor” section, and submit your "Accept" recommendation.

Reviewer #1: All comments have been addressed

Reviewer #2: All comments have been addressed

2. Is the manuscript technically sound, and do the data support the conclusions?

Reviewer #1: Yes

Reviewer #2: Yes

3. Has the statistical analysis been performed appropriately and rigorously? 

Reviewer #1: Yes

Reviewer #2: N/A

4. Have the authors made all data underlying the findings in their manuscript fully available?

Reviewer #1: Yes

Reviewer #2: Yes

5. Is the manuscript presented in an intelligible fashion and written in standard English?

Reviewer #1: Yes

Reviewer #2: Yes

6. Review Comments to the Author

Reviewer #1: The authors have diligently corrected the manuscript and I am happy to recommend it for publication in PLOSone.

Reviewer #2: The authors have adequately addressed all comments. From my perspective, there are no additional changes necessary.

7. PLOS authors have the option to publish the peer review history of their article (what does this mean?). If published, this will include your full peer review and any attached files.

Reviewer #1: No

Reviewer #2: No

---

## [Editor Report · Acceptance letter]

7 Nov 2023

PONE-D-23-28318R1 

Strategies for the production of isotopically labelled Fab fragments of therapeutic antibodies in *Komagataella phaffii* (*Pichia pastoris*) and *Escherichia coli* for NMR studies 

Dear Dr. Aubin:

I'm pleased to inform you that your manuscript has been deemed suitable for publication in PLOS ONE. Congratulations! Your manuscript is now with our production department. 

Kind regards, 

on behalf of

Dr. Rui Tada 

Academic Editor

PLOS ONE